# Health of mothers of children with a life-limiting condition: a protocol for comparative cohort study using the Clinical Practice Research Datalink

Lorna Katharine Fraser ,[1] Fliss E M Murtagh,[2] Trevor Sheldon,[1] Simon Gilbody,[1] Catherine Hewitt[1]

[1]Health Sciences, University of York, York, UK
[2]Wolfson Palliative Care Research Centre, Hull York Medical School, University of Hull, Hull, UK

**Correspondence to**
Dr Lorna Katharine Fraser; lorna.fraser@york.ac.uk

## ABSTRACT

**Introduction** There are now nearly 50 000 children with a life-limiting or life-threatening condition in the UK. These include conditions where there is no reasonable hope of cure and from which they will die, as well as conditions for which curative treatment may be feasible but can fail, for example, cancer or heart failure. Having a child with a life-limiting condition involves being a coordinator and provider of healthcare in addition to the responsibilities and pressures of parenting a child who is expected to die young. This adversely affects the health and well-being of these mothers and affects their ability to care for their child, but the extent of the impact is poorly understood. This study aims to quantify the incidence and nature of mental and physical morbidity in mothers of children with a life-limiting condition, their healthcare use and to assess whether there is a relationship between the health of the mother and the child's condition.

**Methods and analysis** A comparative cohort study using data from the Clinical Practice Research Datalink and linked hospital data will include three groups of children and their mothers (those with a life-limiting condition, those with a chronic condition and those with no long-term health condition total=20 000 mother–child dyads). Incidence rates and incidence rate ratios will be used to quantify and compare the outcomes between groups with multivariable regression modelling used to assess the relationship between the child's disease trajectory and mother's health.

**Ethics and dissemination** This study protocol has approval from the Independent Scientific Advisory Committee for the UK Medicines and Healthcare products Regulatory Agency Database Research. The results of this study will be reported according to the STROBE and RECORD guidelines. There will also be a lay summary for parents which will be available to download from the Martin House Research Centre website (www.york.ac.uk/mhrc).

### Strengths and limitations of this study

► This study is utilising a large, longitudinal, nationally representative primary care data source.
► The Clinical Practice Research Datalink (CPRD) data enables the identification of mother and child dyads and linkage to secondary care data.
► This study is reliant on identification of the cohort of interest by diagnostic coding; therefore, on the quality of diagnostic coding within the CPRD and Hospital Episodes Statistics datasets.

which there is no reasonable hope of cure and from which children or young people will die, as well as conditions for which curative treatment may be feasible but can fail, such as cancer or heart failure. In children and young people, more than 300 diagnoses are life-limiting or life-threatening,[2] including Duchenne muscular dystrophy, severe cerebral palsy, neurodegenerative conditions and severe congenital anomalies.

Although many of the individual diagnoses are rare, collectively life-limiting conditions affect more children and young people than more common single long-term conditions, such as diabetes mellitus.[3] Many of these children are living longer due to the use of medical technologies, for example, ventilation and gastrostomy feeding, and more aggressive treatment of complications and they are often high users of healthcare services.[4] The parents of these children have an added layer of 'pressure' from having a child with a chronic condition who also has a shortened life expectancy.[5] Having a child with a life-limiting condition often involves being a coordinator and provider of healthcare as well as being a parent, and this responsibility exists 24 hours a day, 7 days a week, especially for mothers. The health of these mothers is important, both for their own

## INTRODUCTION

The number of children with a life-limiting or life-threatening condition has been rising, with latest figures estimating 49 000 children and young people with a life-limiting condition in the UK.[1] These include conditions for

well-being, and also in terms of their ability to care for their child. Currently the National Health Service and statutory services for these children rarely extend to providing care for the parents. The focus is on mothers rather than fathers in this study as mothers are the main carer for the majority of these children.

The need for further research on the emotional and psychological support and interventions for parents or carers of children with a life-limiting condition was highlighted recently in National Institute for Health and Clinical Excellence (NICE) guidelines on end of life care in children and young people.[6] This stated that research was needed on 'What emotional support do children and young people with a life-limiting condition and their parents or carers need, and how would they like these needs to be addressed?' The NICE guidelines[6] also noted that no studies had quantified the psychological/mental health of mothers of children with a life-limiting condition. There have been some attempts to quantify these in children with special needs[7] or specific disabilities,[8 9] which have shown higher levels of parental distress or emotional problems than parents of healthy children (36% cf 20%). These, however, do not address the specific needs of those with a life-limiting condition or the added burden that parents of children with a life-limiting condition face, knowing their child is likely to die.

Evidence about the physical health of mothers with a life-limiting condition is also sparse, although two studies in mothers of children with disabilities found a higher prevalence of physical conditions compared with mothers of healthy children. For example, the prevalence of back pain and hypertension was 35.2% and 24.7%, respectively, in mothers of children with disabilities, compared with 26.7% and 19.1% in mothers of healthy children. However, these studies were cross-sectional, self-reported and were not explicitly for mothers of children with a life-limiting condition.[9 10] To date, no study using a nationally representative sample has assessed the incidence or prevalence of mental and physical health conditions in mothers of children with a life-limiting condition. Quantifying the extent and nature of both physical and mental health conditions in these mothers is important when trying to develop and target appropriate preventative or treatment interventions for this population.

This study aims to quantify the incidence and nature of mental and physical morbidity in mothers of children with a life-limiting condition, their healthcare use and to assess whether there is a relationship between the health of the mother and the child's condition.

## METHODS AND ANALYSIS
### Research questions
1. What is the nature and incidence of mental and physical morbidity in mothers of children with a life-limiting condition?
2. What is the relationship between the health of the mother and the child's condition?
3. How many primary care visits do these mothers have compared with other mothers?
4. How many hospital admissions do these mothers have compared with other mothers?
5. What are the resource costs of healthcare for these mothers?

### Data sources and sample selection
This will be an observational comparative cohort study design using data from the Clinical Practice Research Datalink (CPRD). The CPRD dataset contains anonymised, longitudinal records of primary care from a representative sample of general practitioner (GP) practices across the UK (covering approximately 8.5% of the UK population).[11] Individuals within the primary care CPRD dataset are also linked to secondary care data (inpatient, outpatient, A&E Hospital Episodes Statistics (HES) and the Mental Health Minimum Dataset (MHMDS))[12 13] and Office for National Statistics death certificate data. The CPRD provides a unique opportunity to assess the relationship between maternal morbidity and their child health due to the longitudinal nature of data collection (since 1987) and the ability to link mothers and children via their mother–baby link algorithm.[14] It is not currently possible to link babies to fathers in this dataset.

The index children will be extracted from the CPRD mother–baby link, if both of the following criteria are met: 1. the mother has at least 1 year registration 2. they are eligible for HES linkage (resident in England). Children will be grouped into those with a life-limiting condition (n=5000) or chronic condition that is not life-limiting, for example, diabetes, asthma (n=5000) and they will be identified within the CPRD dataset between 2007 and 2017, using Read codes in the CPRD data and ICD-10 codes in the HES data[1] (see table 1) . The development of these ICD-10 code list has been described previously for life-limiting[1] and chronic conditions.[15] Subjects will be matched by year of birth, sex and region to the control group of healthy children (n=10 000) with no long-term conditions.

All primary and the linked secondary care data will be extracted for each mother and child dyad.

### Sample size
A sample size calculation based on 80% power, 5% significance to detect an incidence rate ratio of 1.4 (with a mean incidence rate of 0.03 in maternal anxiety/depression) indicated a minimum of 3260 participants required in each group. The final cohort includes all children with a life-limiting condition in the CPRD dataset. The cohort for analyses contained a total of 35 683 mothers, of whom 8950 had a child with a life-limiting condition, 8868 had a child with a chronic condition and 17 865 had a child with no long-term condition.

### Exposures
The key exposure is the disease status of the child (life-limiting condition/chronic disease (asthma or diabetes)/

**Table 1** Cohort identification processes

| Groups | Inclusion criteria | Exclusion criteria |
|---|---|---|
| 1. Life-limiting or life-threatening condition | ▶ Children from the source population who have prevalent diagnosis event records for life-limiting conditions, either in the Clinical or Referral files in CPRD GOLD based on the Read codes detailed in online supplementary material, or in HES APC based on the ICD-10 codes in online supplementary material.<br>▶ Children have the above events before the end of the study period (31 December 2017).<br>▶ Children have the above events within their UTS follow-up period.<br>▶ Children are aged 18 years or less on the diagnosis event date. | None |
| 2. A chronic condition that is not life-limiting | One matching control will be provided for each case in Group 1. The controls will comprise patients from the source population who fulfil the following criteria:<br>▶ Children from the source population who have diagnosis event records for other chronic conditions in the Clinical or Referral files in CPRD GOLD based on the Read codes detailed in online supplementary material, or in HES APC based on the ICD-10 codes in online supplementary material.<br>▶ Children have the above events before the end of the study period (31 December 2017).<br>▶ Children have the above events within their UTS follow-up period.<br>▶ Children are aged 18 years or less on the diagnosis date. | ▶ Children who have deregistered from the CPRD on or before 01 April 2007.<br>▶ Children who are siblings of children in the case population (Group 1), based on having a link to the same mother in the mother–baby link.<br>▶ Children with a diagnosis on or before 31 December 2017 of life-limiting conditions, either in the Clinical or Referral files in CPRD GOLD based on the Read codes detailed in online supplementary appendix 1, or in HES APC based on the ICD-10 codes in online supplementry appendix 2. |
| 3. No long-term conditions | Up to two matching controls will be provided for each case in Group 1. The controls will comprise of patients from the source population who fulfil the following criteria:<br>▶ Children from the source population who have at least 1 day of registration during follow-up start and end. | ▶ Children with a diagnosis of life-limiting conditions on or before 31 December 2017, either in the Clinical or Referral files in CPRD GOLD based on the Read codes detailed in online supplementary appendix 1, or in HES APC based on the ICD-10 codes in online supplementary appendix 2, will be excluded.Children with a diagnosis of other chronic conditions on or before 31 December 2017, in the Clinical or Referral files in CPRD GOLD based on the Read codes detailed in online supplementary appendix 3, or in HES APC based on the ICD-10 codes in online supplementary appendix 4, will be excluded.<br>▶ Children who have deregistered from the CPRD on or before 01 April 2007.<br>▶ Children who are siblings of children in the case population (Group 1), based on having a link to the same mother in the mother–baby link. |

APC, admitted patient care; CPRD, Clinical Practice Research Datalink; HES, Hospital Episodes Statistics; UPS, up to standard.

no long-term condition). These groups will be identified using previously developed Read and ICD-10 diagnostic code lists (see online supplementary material). The use of both Read and ICD-10 codes will enable identification of relevant diagnoses from either the primary care records (Read) or hospital admission data (ICD-10).

### Outcomes

The outcomes for the mothers will be recorded in either the primary care data or the linked secondary care datasets. The dates of these outcomes will be assessed in relation to the date of the child's diagnoses.

#### Incidence of mental health and physical health conditions

▶ Maternal mental health diagnoses, including both common mental illness and severe mental illness. These will be identified using diagnostic Read codes and relevant prescription data. A previously developed Read code algorithm[16] will be used. The linked MHMDS will indicate the presence of a more serious mental health condition.

▶ Maternal physical diagnoses, including, for example, obesity, hypertension, musculoskeletal problems and cardiovascular disease. These will be identified using diagnostic Read codes, relevant prescription data, HES data and related biometric data (including blood pressure, body mass index and total cholesterol).

#### The association of maternal health with the child's disease trajectory

▶ The number of A&E attendances, hospital admissions and length of stay for the children (HES data).

#### Healthcare use and resources of the mother

▶ Referrals to other services, especially secondary services for mothers (CPRD and HES data).

▶ Uptake of cervical screening by mothers (CPRD data).

▶ The number of primary care attendances (GP, practice nurse, etc) per year by mothers (CPRD data)

▶ The number and nature of prescribed medication for mothers (prescription data).

▶ The number of A&E attendances, hospital admissions and length of stay for mothers (HES data).

### Confounding

The following clinical and demographical confounders will be considered in the analyses:

▶ Ethnicity, this will be classified according to the census 2011 categories[17] using data from CPRD and HES data.

▶ Pre-existing comorbidities of the mother.

▶ Child's diagnoses: grouped into disease categories.[1]

▶ Socioeconomic data is based on the index of multiple deprivation category based on the local super output area of residence.[18]

▶ Region of residence (recognising that the geographical coverage of CPRD is not uniform across England).

### Statistical methods

All analyses will be undertaken using STATA V. 15.[19] All analyses will adjust for known clinical and demographical confounders of importance (age, comorbidities, socioeconomic status, previous use of primary care services, etc). The left-censored and right-censored nature of these data will be taken into account when undertaking these analyses.

#### Incidence of mental health and physical health conditions

Incidence rates of the physical and mental health conditions will be calculated in each group of mothers by dividing the number of cases in each group by the person-time at risk in each group. Incident cases will be counted only after the diagnoses in the child. A comparison with the incidence of these conditions in the three groups of mothers will be undertaken using incidence rate ratios.

#### The association of maternal health with the child's disease trajectory

The presence of physical or mental health diagnoses in the mother will be the outcome variable of interest in a multivariable model with a key covariate being the presence of prolonged or repeated hospital admission for the child. A prolonged admission is one that lasts >14 days and repeated admissions will be defined as more than two admissions in a 6-month time period.

#### Healthcare use and resources of the mother

Using multivariable Poisson regression modelling for the count of GP visits, referrals and hospital admissions with the key exposure of disease status of the child (life-limiting condition/chronic disease/no long-term condition). Multilevel modelling will be used to account for repeated measures within individuals. For the analyses of healthcare costs, we will use information from Unit Costs of Health and Social Care,[20] the NHS national tariffs[21] and Department of Health Reference Costs.[22]

### Missing data

We expect there to be missing data for some of the key covariates in the data, for example, ethnicity. We will assess the level of missingness across all sources of these variable (HES and CPRD). If appropriate, then we will use multiple imputation techniques[23] to address this in order to avoid the potential bias from only undertaking complete case analyses.

### Patient and public involvement

The views of parents and carers of children with a life-limiting condition have informed the development of this study. Initial ideas about the topic area and planned study were discussed with a group of 10 parents of children with a life-limiting condition at Martin House Children's Hospice. They helped to refine the research question and future stages of this programme of work. The Family Advisory Board of the Martin House Research Centre (www.york.ac.uk/mhrc)

will have involvement in this study with an emphasis on the preparation of the lay summaries and dissemination to parents and children.

## Ethics and dissemination

This study protocol has approval from the Independent Scientific Advisory Committee for the UK Medicines and Healthcare products Regulatory Agency Database Research (protocol 18_313). They have NREC approval for release of pseudonymised data for observational research.

This study will be complete by February 2021.

The results of this study will be reported according to the STROBE and RECORD guidelines (ref). There were also be a lay summary for parents which will be available to download from the Martin House Research Centre website (www.york.ac.uk/mhrc).

**Contributors** LKF had the original idea for this study and wrote the first draft of this manuscript. FEMM, SG, TS and CH contributed to the development of this idea, study design and revised the manuscript. All authors approved the final submitted version of this manuscript.

**Funding** This paper is independent research arising from a Career Development Fellowship held by LFK (CDF-2018-11-ST2-002) supported by the National Institute for Health Research. The views expressed in this publication are those of the author(s) and not necessarily those of the NHS, the National Institute for Health Research or the Department of Health.

**Competing interests** None declared.

**Patient and public involvement** Patients and/or the public were involved in the design, or conduct, or reporting, or dissemination plans of this research. Refer to the Methods section for further details.

**Patient consent for publication** Not required.

**Provenance and peer review** Not commissioned; externally peer reviewed.

**ORCID iD**

Lorna Katharine Fraser http://orcid.org/0000-0002-1360-4191

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
