## [Reviewer comments · BMJ Open]

ARTICLE DETAILS

TITLE (PROVISIONAL)	The health of mothers of children with a life-limiting condition; a protocol for comparative cohort study using the Clinical Practice Research Datalink
AUTHORS	Fraser, Lorna; Murtagh, Fliss; Sheldon, Trevor; Gilbody, Simon; Hewitt, Catherine

VERSION 1 - REVIEW

REVIEWER	Holly Hope University of Manchester, UK
REVIEW RETURNED	05-Feb-2020

GENERAL COMMENTS	This is a very interesting study. The impact of child illness on parents is of clinical concern and this study exploits primary care register data and its linkages with MI and HES data-sets to thoroughly investigate this question. One element of the planned analyses that might be clearer are the exposure periods for different mental and nonmental morbidity of the mother, will the direction/proximity of effect be addressed or is this a descriptive study? I also think that effects of the hospital the child is treated in and the gp may need to be taken account in the planned analyses. Prior research suggests quality of care of the child has an impact on the well being of the parent, I would suggest given that there are regional variables available to examine if there is an effect modification by region that indicates areas of the country where families might be particularly struggling. Similarly analyses stratified (rather than adjusted for) by age of child at diagnosis and imd quintile may reveal important moderated effects, for example younger children and those from the poorest areas are likely to receive more admissions to hospital and intensive treatments, and their mothers are at increased risk of mental illness, so the relationship between child illness and maternal health may be different in these groups. You are not addressing the impact on paternal health, which is a limitation, is there a particular reason for this? Will you take account of sibling morbidity in your matched analyses, eg two children with cystic fibrosis is much more challenging than
--

	one child. Overall I think this is a very important study.
--	---

VERSION 1 – AUTHOR RESPONSE

Reviewer(s)' Comments to Author:

Reviewer: 1

Reviewer Name: Holly Hope

Institution and Country: University of Manchester, UK

Please state any competing interests or state 'None declared': None

Please leave your comments for the authors below

This is a very interesting study. The impact of child illness on parents is of clinical concern and this study exploits primary care register data and its linkages with MI and HES data-sets to thoroughly investigate this question.

Thank you for your helpful review.

One element of the planned analyses that might be clearer are the exposure periods for different mental and nonmental morbidity of the mother, will the direction/proximity of effect be addressed or is this a descriptive study?

Incident cases of physical or mental health morbidity in the mothers will only be included after the diagnoses in the child. This has been clarified on page 11.

I also think that effects of the hospital the child is treated in and the gp may need to be taken account in the planned analyses.

An assessment of clustering of effects by GP practice will be undertaken but there will be very few cases per practice.

Prior research suggests quality of care of the child has an impact on the well being of the parent, I would suggest given that there are regional variables available to examine if there is an effect modification by region that indicates areas of the country where families might be particularly struggling.

Variation by region will be assessed in the descriptive statistics and if appropriate will be accounted for in the analyses.

Similarly analyses stratified (rather than adjusted for) by age of child at diagnosis and imd quintile may reveal important moderated effects, for example younger children and those from the poorest areas are likely to receive more admissions to hospital and intensive treatments, and their mothers are at increased risk of mental illness, so the relationship between child illness and maternal health may be different in these groups.

This is an important question although there is an argument that it is not only the age at diagnoses of the child that is a risk factor for the mother's health but also how long since the child has been diagnosed. Stratification by deprivation category may be undertaken but this study was not powered for stratified analyses.

You are not addressing the impact on paternal health, which is a limitation, is there a particular reason for this?

The mother-child link is via pregnancy records therefore is not currently possible within the CPRD to link children with their fathers

Will you take account of sibling morbidity in your matched analyses, e.g. two children with cystic fibrosis is much more challenging than one child.

Yes although there will be very few such examples within these data.

Overall I think this is a very important study.

Thank you